# Overwintering of Thrips (Thysanoptera) Under the Bark of the Plane Tree (*Platanus* x *hispanica* Mill. ex Münchh.) in Southeastern Poland

**DOI:** 10.3390/insects16010092

**Published:** 2025-01-17

**Authors:** Halina Kucharczyk, Marek Kucharczyk, Tomasz Olbrycht

**Affiliations:** 1Department of Zoology and Nature Protection, Institute of Biological Sciences, Faculty of Biology and Biotechnology, Maria Curie-Sklodowska University, 19 Akademicka Str., 20-033 Lublin, Poland; marek.kucharczyk@mail.umcs.pl; 2Department of Agroecology and Forest Utilization, University of Rzeszów, 1a M.Ćwiklińskiej Str., 35-601 Rzeszów, Poland; tkolbr@ur.edu.pl

**Keywords:** thrips, overwintering, London plane, plane tree, tree bark, biodiversity

## Abstract

The study conducted in southeastern Poland investigated the diversity of thrips species overwintering under the bark of the plane tree (*Platanus* x *hispanica*). The study was carried out in 29 sites, including urban and rural parks, over three winter seasons from 2014 to 2016. A total of 494 thrips individuals were collected. Fifteen different thrips species were identified, with ten species being dendrophilous and four belonging to the fungivorous suborder Tubulifera. The most common species found were *Dendrothrips degeeri*, *Phlaeothrips coriaceus*, *Thrips major*, and *Dendrothrips ornatus*. *D. degeeri* was found in the highest number of sites (22), while six species were only found in one site each. The study also revealed a significant correlation between the species composition of thrips collected under the plane tree bark and such factors as insolation and the diameter of the tree trunk where the species overwintered.

## 1. Introduction

The order Thysanoptera (about 6400 known species) includes small and dorsoventrally flattened body insects with a rasping and sucking mouth apparatus, unique due to their lack of a left mandible and wings surrounded by fine hairs. They are divided into two suborders, Tubulifera (3600 species) and Terebrantia (2800 species) [1,2]. Insects of the former suborder belong to only one family, Phlaeothripidae, but the latter suborder includes eight families. In Poland, species of the suborder Terebrantia belong to three families: the Aeolothripidae, Melanthripidae, and Thripidae, the last being the richest in species. The family Aeolthripidae mainly comprises zoophagous species. The other two families contain phytophagous species that feed and reproduce on the leaves and flowers of herbaceous plants and trees. Most tubuliferan species are mycophages feeding on fungal spores, either consuming whole spores or imbibing their sap or fungal hyphae, with the exception of the most numerous genus *Haplothrips*, which is dominated by phytophagous species. In thrips development, five (Terebrantia) or six (Tubulifera) stages are distinguished: egg, two active feeding instars of larvae, inactive non-feeding stages for propupa and pupa (or two pupae in Tubulifera), and adults—a female and a male. Most thrips species reproduce bisexually. The sex determination is haplodiploid; females develop from fertilized eggs and are always diploid forms, and haploid males come from unfertilized eggs. In some species, obligatory thelytokous parthenogenesis occurs. In such cases, only females are observed in populations [3,4].

Arthropods, including insects, have developed different strategies of overwintering in the temperate climate [5,6]. In most thrips species, the adult female stage is dominant, lives longer, and most often overwinters. It can survive unfavorable times in the soil, litter, and dry remnants of their host plants. The choice of the habitat in which the insects overwinter can be critical to their survival. Some species, especially those with two generations per year, may hibernate in the stage of larva or pupa; in this situation, they spend this phase of development mainly in the soil, going down to a depth from a few up to 100 cm, depending on the type of soil. The mortality of thrips is higher when immature stages overwinter, especially in the clay soil [7,8,9]. Species introduced outside their natural range, e.g., invasive pests depending on their tolerance of low temperatures, survive unfavorable climatic conditions in the soil, under leaf litter, and debris of their plant hosts or in greenhouses looking for the best suitable overwintering sites near their hosts. Much more frequently studied is the phenomenon of hibernation and tolerance to low temperatures in thrips species that are crucial agricultural plant pests and virus vectors, e.g., *Thrips tabaci* Lindeman, *Frankliniella occidentalis* (Trybom), *Thrips palmi* Karny, and *Limothrips cerealium* Haliday (Thysanoptera, Thripidae) [7,8,9,10,11,12,13]. In turn, the knowledge about the overwintering of thrips living in natural habitats or in urban or rural green areas is insufficient. Invertebrates, e.g., insects, mollusks, millipedes, and spiders, living on the ground or under decaying logs as well as fungivorous tubuliferans with their lifespan connected with trees very often overwinter under the bark, near trees, or in rotting trunks on the forest floor. Such biotopes provide them with microhabitats in many cracks and crevices, which can protect them from predators and ensure safety during unfavorable periods with strong winds and low temperatures [7,13,14,15,16].

Trees growing in green areas in urban and rural environments, such as squares, parks, and old alleys, can act as biodiversity centers and a refuge for insects [17,18,19]. This is confirmed by the presence of rare species and those protected in Poland, such as *Osmoderma eremita* (Scopoli) (Scarabaeidae, Coleoptera) and *Cucujus cinnaberinus* (Scopoli) (Cucujidae and Coleoptera), is such sites [20,21,22]. The management in these areas, e.g., mowing lawns, removing dead trees and fallen branches, and raking litter and fallen dry leaves in autumn, limits the potential wintering places of invertebrates [23,24]. The irregular, loose, and flaky structure of tree trunk bark, as in *Platanus* spp. (Platanaceae), allows small insects to overwinter on the trunks, especially in urban areas. Wintering under such bark provides many advantages for insects, including higher temperatures than on the ground, especially on the sunny side of the trunk, and the ability to aggregate. Moreover, moist bark can be a habitat for fungi which serve as food for mycophagous species. Additionally, in spring, tree trunks warm up faster than the soil, which allows insects to leave their wintering sites earlier. On the other hand, there are higher temperature fluctuations between days and nights, especially in spring, which may be dangerous for insects with soft bodies [5,8,13].

The London plane tree is an artificial horticultural hybrid between the North American plane tree (*P. occidentalis* L.) and the Oriental plane tree (*P. orientalis* L.). The former species is native to North America, and the latter is indigenous to the Mediterranean region of Europe and Asia. The hybrid between both these species was known under the name *Platanus* x *acerifolia* (Aiton) Willd., but its valid name after revision is *P.* x *hispanica* Mill. ex Münchh. Outside its natural range, *P.* x *hispanica* was noticed before 1700 in England in the Oxford Botanical Garden. The name sycamore is used for the *P. occidentalis* in America, but the hybrid is named London plane or plane tree [25,26]. Due to the enormous size of old trees, their ornamental value, and their resistance to air pollution and pests, this tree is recommended for use as a buffer in city parking lots or roadsides and as a decorative element in green areas or along city avenues [26]. Since the XVIII century, the London plane has been planted successfully in city parks, rural residences, and along roads in Poland, mostly in the western parts of this country, where the climate is milder than in the eastern part [27]. Due to climate change, including milder winters and higher spring temperatures, different varieties of this tree are increasingly being planted successfully in eastern Poland. Planting this tree outside its natural range causes the appearance of alien and invasive species, e.g., the sycamore lace bug *Corythucha ciliata* (Say) (Hemiptera:Tingidae) and the plane tree bug *Arocatus longiceps* Stål (Hemiptera: Lygaeidae), in new areas [28]. So far, surveys of bugs and spiders overwintering under the plane tree bark have been carried out in Poland, also to determine the presence of pests of this tree [29,30,31,32,33]. However, thrips associated with the plane tree (*Platanus* x *hispanica*), especially those overwintering on this tree, have not been studied.

The aims of the study were (1) to identify thrips species overwintering under plane tree bark and (2) to find a correlation between species diversity and insect abundance and such factors as the trunk circumference, the number of branches diverging from a common trunk, and the light conditions in which the studied trees grew.

## 2. Materials and Methods

### 2.1. Study Area

The thrips specimens were collected during the realization of a project focused on a search for bugs (Hemiptera:Heteroptera) overwintering under the bark of plane trees [31]. Samples of the thrips species to be analyzed were found in 29 sites in southeastern Poland; 23 sites were located in the Sandomierz Basin, five in the Eastern Beskid Mountains, and one in the Roztocze region, mainly in Podkarpackie and Małopolskie Provinces. The trees were located in urban areas, mainly in city parks and rural parks surrounding palaces, castles, and noble manors established in the XVII-XIX century in southeastern Poland (Figure 1, Appendix A). Most of them were landscaped parks in the English style with large lawns, in the center of which exotic trees, such as plane trees, tulip trees, black walnut, and foreign conifers, were planted. After the Second World War, the residences were nationalized. About half of the studied sites are in private hands now, and several of the residences are used as museums (Krasiczyn, Tarnów, and Żarnowiec), schools, or hotels (Dubiecko). In the studied sites, the plane trees form groups of several individuals, alleys, or most often grow singly due to their decorative value.
Figure 1Location of the study sites in southeastern Poland: BA—Bachórzec, BO—Bolestraszyce, CZ—Czudec, DU—Dubiecko, Dz—Dzikowiec, JA—Jasionka, KL—Klemensów, KO—Kombornia, KR—Krasiczyn, LO—Łopuszka Mała, ME—Medyka, MP—Miejsce Piastowe, NI—Nisko, PE—Pełkinie, PR—Przemyśl, PW—Przeworsk, RO—Roźwienica, RZ—Rzeszów, SL—Słocina, TA—Tarnów, UR—Urzejowice, WE—Werynia, WZ—Wzdów, WS—Wola Sękowa, ZA—Zaczernie, ZS—Zasów, ZW—Zawada, ZR—Żarnowiec, and ZU—Żulin. The blue square marks the part of Poland where the research was conducted.
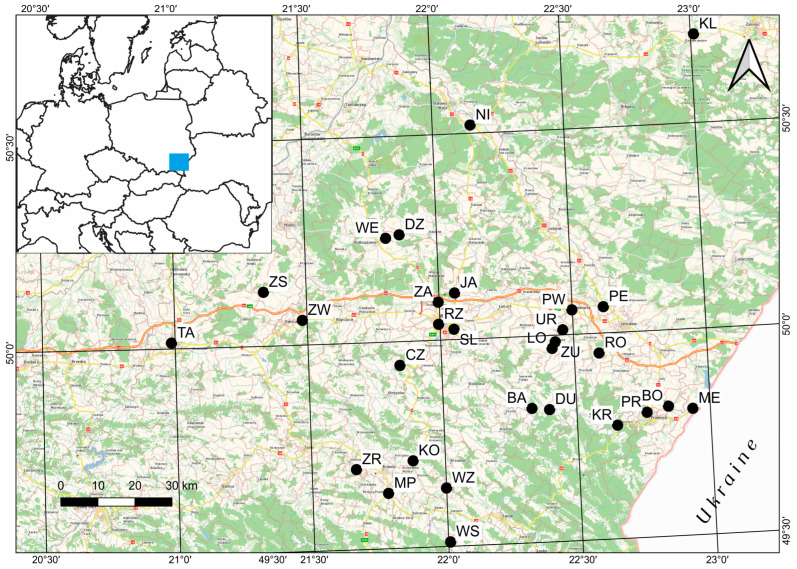


### 2.2. Sampling Methods

The research was conducted between February and April 2014–2016. A total of 29 samples were taken, 25 in 2014, three in 2015, and one in 2016 (Appendix A). Before leaves developed, the bark of the trees was collected in plastic bags with a volume of 4 L. From each trunk, one bark sample (about 250 cm^2^) was taken at a height of 1–2 m (Figure 2). Using a thin brush, the thrips specimens were removed from the thinner pieces of the bark in the laboratory under a stereomicroscope OLYMPUS SZ51 (Olympus Corporation, Tokyo, Japan) and kept in 70% ethyl alcohol. Due to the small size of the insects, microscopic slides had to be prepared for species identification. The slides were prepared as in Mound and Kibby [34], and the specimens were identified under the OLYMPUS BX 60 microscope (Olympus Corporation, Tokyo, Japan) using keys developed by zur Strassen [35] and Schliephake and Klimt [36]. The species photographs were taken by the first author with an OLYMPUS DP 72 camera (Olympus Corporation, Tokyo, Japan) (Figure 2). Based on the information provided by the two keys, the food preference and geographic range of all the species found in the samples were characterized (Table 1). The specimens are deposited in the Thysanoptera collection of the Department of Zoology and Nature Protection, Maria Curie-Skłodowska University, Lublin, Poland.

### 2.3. Environmental Data

To investigate the environmental effects on the diversity of an overwintering thrips community, we considered three explanatory variables: (i) the trunk circumference (in meters) at a height of 1.5 m, (ii) the number of trees in every site or the number of trunk branches at a height of 1.5 m, and (iii) the insolation of the tree trunks. The insolation of the tree trunks was estimated based on the direction from which the trunk was best illuminated. The aspect azimuth was converted from the 0° to 360° compass scale to a value between 0 and 1 (Transformation of Aspect, TRASP; [37]). The conversion was accomplished using the following equation:$$TRASP=\frac{1-\cos\left[\left(\pi /180\right)\left(aspect-30\right)\right]\;}{2}$$

This transformation assigned a value of one to hotter and dryer south–southwesterly illuminated trunks and trees that grow in full light, and a value of zero was assigned to trunks illuminated from the north–northeast direction (typically the coolest and wettest orientation) or fully shaded trees. In sites where samples were obtained from more than one tree, the mean TRASP index was calculated.

### 2.4. Data Analysis

The material was analyzed with respect to the species and the ecological (functional) group, i.e., dendrophilous thrips, using the faunistic quantitative similarity (the Bray–Curtis formula and diversity index (Simpson (1 − D) formula) [38].

Non-metric multidimensional scaling (NMDS) ordination analysis in the PAST 4.17c package [39] was used to identify similarity in the data set and to detect whether there was any potential environmental gradient impinging on the wintering thrips fauna. We used the raw species abundance data from each study site and the Sørensen index (Bray–Curtis distance).

In addition, we used the raw species abundance data from each study site for the construction of Generalized Linear Model (GLM) assuming a Poisson distribution and using a logarithm as a link function. To determine which of these explanatory variables were the most decisive in the community structure, we used an information theoretical approach based on the Second-Order Akaike Information Criterion (AICc), which is indicated for small sample sizes, and the best model was indicated by the lower AICc value [40]. The model suitability was assessed by the homogeneity and normality of the residuals. GLMs were performed using package glm2 in the R software v.4.4.2 [41].

Multivariate ordination analyses were used to determine the environmental parameters responsible for the abundance and composition of wintering thrips. Detrended Correspondence Analysis (DCA) was used first to detect the gradient length and, since the gradient was short (<3 SD), the linear method, (Redundancy Analysis (RDA)), was applied. Two separate analyses were applied to all the species and to the dendrophilous species only. To test the significance of the variables (*p* < 0.05), the forward selection procedure was used with the Monte Carlo permutation test. RDAs were carried out in CANOCO 4.5 for Windows [42].

## 3. Results

Our research has shown that the trunks of plane trees (*P.* x *hispanica*) growing in anthropogenically modified areas, e.g., big city parks and parks surrounding manor houses, provided a suitable habitat for thrips overwintering. During the research conducted in 2014–16, 15 species (nine of Terebrantia and six of Tubulifera orders) belonged to 11 genera and two families (Thripidae and Phlaeothripidae), and 494 specimens of adult thrips were found. 84% (415 specimens) of all the individuals collected in the study belonged to the suborder Terebrantia. The most numerous and most frequently collected were *Dendrothrips degeeri* (68.2%), followed by *Phlaeothrips coriaceus* (11.7%), *Thrips major* (6.5%), and *Dendrothrips ornatus* (6.3%). The other species were selected individually. Among the species noted, females predominated (478 ind., 96.8%), while only a few males were found (16 ind., 3.2%). The exception was *Ph. coriaceus*, whose males constituted 21% of all the specimens found. The dominant *D. degeeri* was found under the bark collected in 22 of the 29 studied sites, followed by *P. coriaceus* (11 sites) and *D. ornatus* (10 sites). The number of thrips species recorded varied from site to site, with the greatest number—six species—found under the bark of plane trees growing in the arboretum in Bolestraszyce, where the trees are surrounded by other exotic and native plants, and in Dzikowiec and Dubiecko parks, where the trees grow singly on a mown lawn (Figure 2). Despite the highest number of species recorded in the aforementioned sites, the highest number of individuals was collected in Dzikowiec (111 ind.), Łopuszka Mała (90 ind.), and Medyka (73 ind.). In all the sites, *D. degeeri* was the dominant species, followed by *Ph. coriaceus* in Dzikowiec.

Among the taxa found, ten species were dendrophilous, of which five of the tubuliferans were fungivorous and one was classified as a predator. Four terebrantian dendrophilous species were foliophags feeding on leaf blades of deciduous trees. In four sites, only one species, *M. consociatus*, *M. salicis*, *P. denticauda*, or *P. dianthi* was found; all except the last one are dendrophilous. The tubuliferan species *A. nodicornis*, *H. bidens*, *H. williamsianus*, and *Ph. denticauda* have been reported from a few localities in Poland so far. In terms of geographical distribution, six of the species found during the research have the European range, three have the Euro-Siberian occurrence range, and six are Palearctic species. Several of the species were introduced into North America, Asia, New Zealand, and Australia (Table 1, Figure 2) [35,36].
insects-16-00092-t001_Table 1Table 1List and characteristics of thrips species overwintering under the plane tree bark.SpeciesAcronymNo. of SpecimensCharacteristicssuborder Terebrantia*Dendrothrips degeeri* Uzel 1895*De_d*337 ♀♀D, Fol (*Fraxinus* and other deciduous trees), E*Dendrothrips ornatus* (Jablonowski 1894)*De_o*31 ♀♀D, Fol (*Fraxinus, Ligustrum, Syringa, Corylus, Tilia*), P, introduced in North America*Frankliniella intonsa* (Trybom 1895)*Fr_i*3 ♀♀Pol, Flor (herbaceous plants), P, introduced in Canada and southeastern Asia*Limothrips denticornis* Haliday 1836*Li_d*2 ♀♀Pol, G (leaf blades of various grass species), E–S, introduced in North America and Australia*Mycterothrips consociatus* (Targioni-Tozzetti 1886)*My_c*1 ♀D, Fol (*Alnus, Betula, Corylus, Quercus, Salix*), E–S*Mycterothrips salicis* (O.M. Reuter 1879)*My_s*1 ♀D, Fol (*Populus, Salix*, less often *Alnus* and *Tilia*), E-S, introduced in North America*Pezothrips dianthi* (Priesner 1921)*Pe_d*1 ♀Mon, Flor (*Dianthus carthusianorum*), W-P*Thrips fuscipennis* Haliday 1836*Th_f*6 ♀♀Pol, Flor (herbaceous plants, preferred Rosaceae), P, introduced in North America*Thrips major* Uzel 1895*Th_m*32 ♀♀, 1 ♂Pol, Flor (herbaceous plants and *Sambucus nigra*), P**suborder Tubulifera***Acanthothrips nodicornis* (O.M. Reuter 1880)*Ac_n*4 ♀♀D, F (under bark of dead deciduous trees), P*Hoplandrothrips bidens* (Bagnall 1910)*Ho_b*3 ♀♀, 1♂D, F (under bark of dead deciduous trees), E, introduced in New Zealand*Hoplandrothrips williamsianus* Priesner 1923*Ho_w*3 ♀♀D, F (dry branches of *Salix*),*Phlaeothrips coriaceus* Haliday 1836*Ph_c*46 ♀♀, 12 ♂♂D, F (on branches and under bark of deciduous trees), E and North America*Phlaeothrips denticauda* Priesner 1914*Ph_d*1 ♀D, F, Hyg (on branches and under bark of *Salix* and *Alnus*), E*Xylaplothrips fuliginosus* (Schille 1911)*Xy_f*7 ♀♀, 2 ♂♂D, Pred (bark and leaves of trees), EExplanations: D—dendrophilous, F—fungivorous, Flor—floricolous, Fol—foliicolous, G—graminicolous, Mon—monophagous, Pol—polyphagous, Pred—predatory, Hyg—hygrophilous; geographical range: E—European, E–S—Eurosiberian, P—Palearctic, and W–P—West-Palearctic. Acronyms used in Figure 3 and Figure 4.
Figure 2Most numerous collected species (**a**–**e**) and example of study site (**f**,**g**). (**a**)—*Dendrothrips degeeri*, (**b**)—*Dendrothrips ornatus*, (**c**)—*Thrips major*, (**d**)—*Phlaeothrips coriaceus*, (**e**)—*Xylaplothrips fuliginosus*; (**f**)—plane tree bark with traces of the sample taken, and (**g**)—singly growing plane tree in Dubiecko (Original photos by H. Kucharczyk).
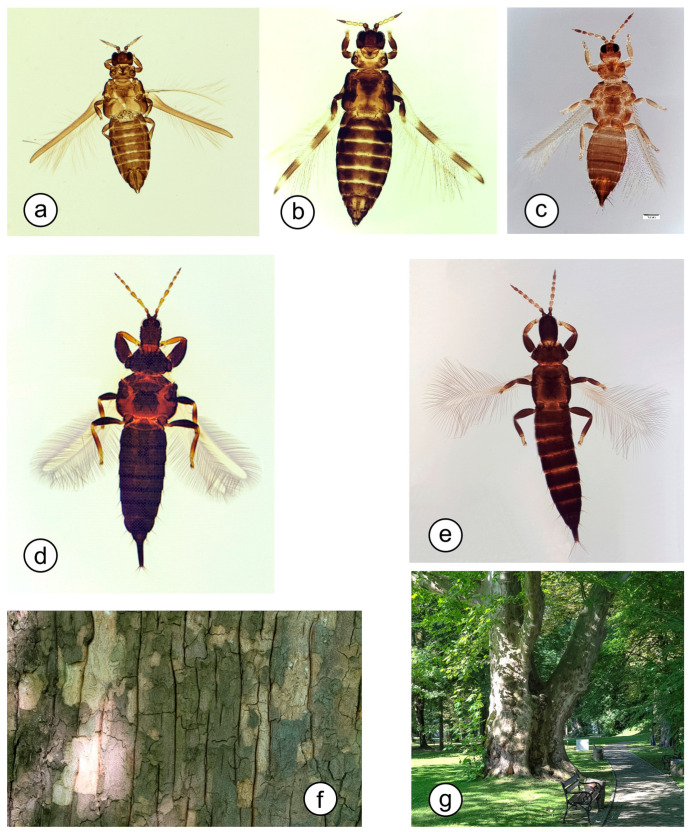



The non-metric multidimensional scaling (NMDS) approach indicated that the four sites clearly differed in their species composition and abundance from the others (Figure 3a). These were Medyka (ME), Łopuszka Mała (LO), Dzikowiec (DZ), and Klementowice (KL). The first three were distinguished by the number of wintering *D. degeeri* (also *Ph. coriaceus* in DZ), and at KL (the most northern site located in Roztocze region) was characterized by the presence of numerous *Thrips major* individuals. The isolated position of such sites as Dubiecko (DU), Słocina (SL), and Zawada (ZW) on the left side of the graph was characterized by a higher number of *D. ornatus* in the first site and *D. degeeri* in the other two sites, whereas Wzdów (WZ) and Czudec (CZ) exhibited a high number of specimens of *Ph. coriaceus*. The distribution of the dendrophilous thrips communities also indicated the distinctiveness of the Medyka (ME), Łopuszka Mała (LO), and Dzikowiec (DZ) sites (Figure 3b). The thrips assemblages in the other sites were homogeneous and consisted of singular specimens.

**Figure 3 insects-16-00092-f003:**
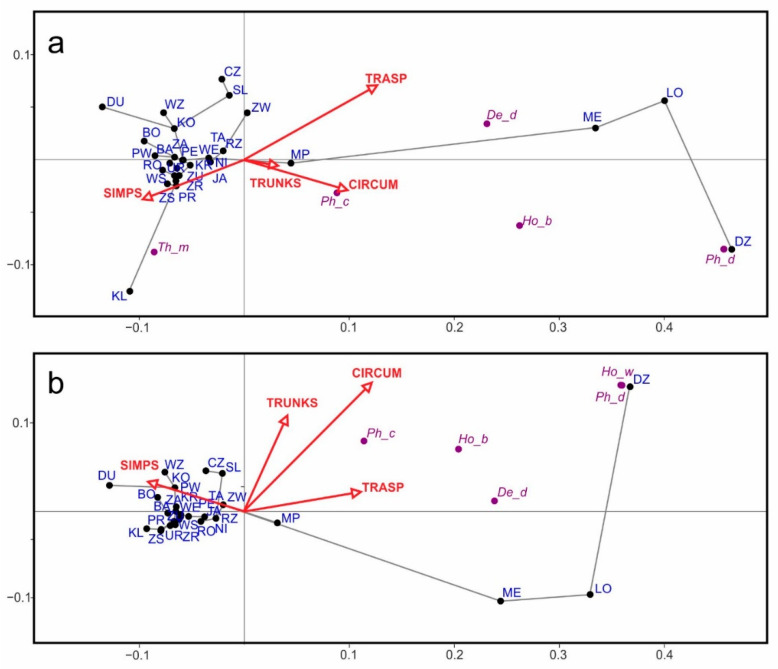
Two-dimensional non-metric multidimensional scaling (NMDS) plot showing the arrangement of the study sites based on Bray–Curtis faunistic similarities: (**a**)—all thrips species and (**b**)—dendrophilous thrips species. Acronyms of the study sites, as shown in Figure 1, and the names of thrips species, as shown in Table 1. For readability, only selected species are shown in the plot.

The abundances of all overwintering thrips and dendrophilous thrips were positively correlated only with the insolation index (Spearman Rho = 0.615 *p* < 0.001 and Rho = 0.718 *p* < 0.001, respectively). The circumference and number of trunks in the site were not significantly correlated with the abundance of overwintering thrips. The Simpson diversity index (SIMPS) was not significantly correlated with the environmental factors.

According to the generalized linear model (GLM), the total number of individuals overwintering under the plane tree bark was significantly influenced by the trunk circumference, the number of trees in the site, and the insolation of the tree trunks expressed as TRASP (Table 2). The model was described by the equation:Number of individuals (N) = e^[(1.162+ (0.103 × CIRCUM) − (0.333 × TRUNKS) + (2.169 × TRASP)], 
where N is the total number of individuals.

The model was able to explain 60% of the variance in the avoidance response ([(null deviance − residual deviance)/(null deviance)] × 100).

The number of dendrophilous thrips wintering under the bark was significantly influenced by the same environmental variables as above (Table 3). The model relating to the dendrophiles was described by the equation:Number of individuals (N_de_) = e^[((0.100 × CIRCUM) − (0.340 × TRUNKS) + (3.416 × TRASP)], 
where N_de_ is the number of individuals of dendrophilous species.

The model was able to explain 75% of the variance in the avoidance response.

**Table 2 insects-16-00092-t002:** Model fit of the generalized linear model (using Poisson distribution) and numerical output for the abundance of overwintering thrips (all collected species) to the trunk circumference at a height of 1.5 m (CIRCUM), the number of trees in the site or the number of trunk branches at a height of 1.5 m (TRUNKS), and the insolation of the tree trunks (TRASP).

	Estimate	Standard Error	Z Value	*p*
Intercept	1.162	0.1600	7.260	<0.001
CIRCUM	0.103	0.0100	10.233	<0.001
TRUNKS	−0.333	0.0538	−6.194	<0.001
TRASP	2.169	0.1930	11.239	<0.001

Null deviance: 797.83 on 28 *df.* Residual deviance: 320.41 on 25 *df.* AICc 442.89.

**Table 3 insects-16-00092-t003:** Model fit of the generalized linear model (using Poisson distribution) and the numerical output for the abundance of overwintering dendrophilous thrips to the trunk circumference at a height of 1.5 m (CIRCUM), the number of trees in the site or the number of trunk branches at a height of 1.5 m (TRUNKS), and the insolation of the tree trunks (TRASP).

	Estimate	Standard Error	Z Value	*p*
Intercept	0.047	0.2385	0.198	0.843
CIRCUM	0.100	0.0107	9.346	<0.001
TRUNKS	−0.340	0.0580	−5.857	<0.001
TRASP	3.416	0.2730	12.513	<0.001

Null deviance: 825.21 on 28 *df.* Residual deviance: 208.99 on 25 *df.* AICc 326.06.

The separate redundancy analysis (RDA) of the influence of environmental variables relating to the species composition and abundance showed that these factors significantly explained the overall variability of the wintering thrips (49.3%).

In this analysis (Figure 4a), all environmental variables were statistically significant, in descending order: the trunk circumference (CIRCUM), the insolation of the tree trunks (TRASP), and the number of trees in the site or the number of trunk branches (TRUNKS).

The occurrence of *H. bidens*, *H. williamsianus*, *Ph. coriaceus*, and *Ph. denticauda* was clearly correlated with the size of the trunks (CIRCUM), and the presence of *F. intonsa*, *X. fuliginosus*, and *D. ornatus* was correlated with the number of trees (TRUNKS). The insolation index (TRASP) was clearly negatively correlated with the occurrence of *T. major*.

A separate analysis concerning only dendrophilous species also showed a significant impact of the environmental factors on the species composition and abundance (Figure 4b). These factors significantly explained the variability of this group (50.0%).

The first axis defined the size of trunks (CIRCUM) and the insolation index (TRASP), and the second axis showed the number of trees (TRUNKS). The association of species occurrence indicated that the total number of wintering thrips and the occurrence of *D. degeeri*, *H. bidens*, *H. williamsianus*, *Ph. coriaceus*, and *Ph. denticauda* were correlated with the first two factors. The abundance of *X. fuliginosus* and *D. ornatus* was influenced by the number of trees occurring in the study site (TRUNKS).
Figure 4RDA biplots showing overwintering thrips assemblages in relation to environmental variables, (**a**)—all noted species; (**b**)—only dendrophilous species. The acronyms of the variables and thrips species names are given in Table 2 and Table 3 and in Table 1, respectively.
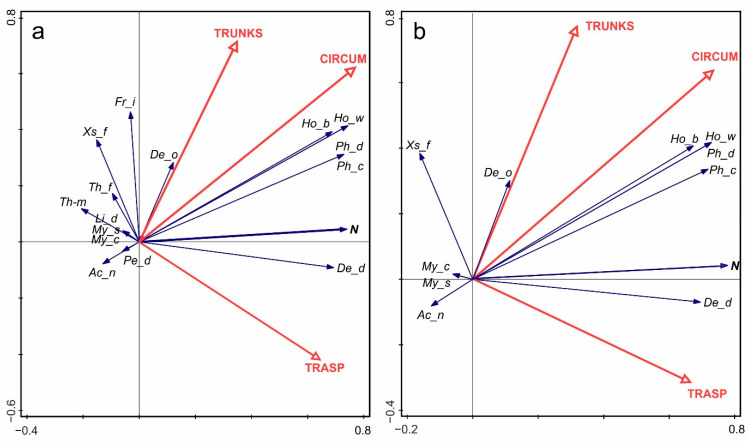



## 4. Discussion

In urban areas, parks with heterogeneous vegetation constitute enclaves characterized by greater biodiversity. In rural areas, manor parks and alleys of old trees have a similar function, increasing native biodiversity in the agricultural landscape [17,18,19,23]. In light of climate change, the expanding ranges of invasive species, and the reduction in acreage and degradation of natural habitats, these areas have become sites for intensive studies of change occurring in both flora and fauna [24,43]. The research conducted in the Rzeszów (Poland) historic parks revealed the presence of several interesting ground and saproxylic beetle species, including those belonging to the Carabidae, Cerambycidae, Elateridae, and Scarabeidae families, which are protected under relevant legislation, e.g., the hermit beetle *Osmoderma eremita* (Scopoli) (Coleoptera: Scarabaeidae), which is an umbrella species listed in Anex II of the EU Habitat Directive [18,44,45,46]. In the reserve “Lisia Góra” and the Zalesie forest in the Rzeszów area, Olbrycht, and Kucharska-Świerszcz [22] have found another strictly protected species, i.e., the bark beetle *Cucujus cinnaberinus* (Scopoli) (Coleoptera: Cucujidae), which indicates the great importance of green areas for preserving biodiversity. In our research, this was confirmed by the finding of *H. williamsianus*, a species which has been found in Poland only in the Białowieża Primeval Forest so far, and *A. nodicornis* and *Ph. denticauda*, which are rarely collected both in Poland and in Europe [47,48,49]. The research on the communities of harvestmen (Opiliones) in the historical park in Rusovce (Slovakia) showed the high diversity of ground-living species in anthropogenically changed ecosystems. In seven sites covered by different management practices, the authors recorded 13 species, seven of which were caught in a plane tree alley [24]. Terrestrial arthropods, e.g., ants, arachnids, and harvestmen, were studied in Warsaw (Poland) in variously managed urban green spaces. Using pitfall traps, the authors distinguished both generalist and specialist species in arthropod communities and showed high levels of this diversity in botanical gardens, urban parks, and urban forests where different management practices were applied. They noticed that the greenery urban areas are refuges for arthropods in the transformed city environment [50].

Studies on arthropods associated with trees during their life are insufficient. Various research methods are used for the identification of arthropod species, i.e., Moericke traps and screen traps, while pitfall traps and arboreal photoeclectors are used for arthropods living on tree bark [47,51,52,53,54,55]. These methods are often used during the full leaf development period, from spring to autumn, to study arthropods living in tree canopies or migrating along trunks from wintering sites in soil or litter. Using various methods, observations on spiders and harvestmen living on the trees were carried out [32,51,56,57,58]. The numerous species of observed arachnids were classified to three groups: accidental, which spent only a short time on the bark, facultative bark-dwellers, which spent a longer time on the bark of trunks or branches, e.g., species wintering in this niche, and the third group of real or exclusive bark-dwellers, which may be found on bark throughout the year [56,57,58]. This classification can also be applied to the thrips species found under the bark of plane trees in our study. Most of them belong to the first and second groups, i.e., polyphagous species feeding and breeding on herbaceous plants represent the first group, while dendrophilous foliophages are representatives of the second one. *D. degeeri*, the most numerous and most common species collected during our research, appears to belong to the second group, too. This species probably not only overwinters on the plane tree but also feeds on its leaves during the growing season. The plane tree has not previously been listed as a host plant for this thrips species. Fungivorous species probably feed on fungi present in humid places under the bark or on dead logs or branches on the ground and may be classified into the third group. These species are highly dispersed and live in hiding. Their relationship with the plane tree needs further research.

Our research showed that 15 species of thrips found a safe place to overwinter under the bark of plane trees. Of these, 13 species were caught in Moericke traps during ecological monitoring surveys in oak-hornbeam forests in the Białowieża Primeval Forest. *D. degeeri*, *D. ornatus*, *M. consociatus*, *M. salicis*, *A. nodicornis*, and *Ph. denticauda* were caught singly, but *H. bidens* and *H. williamsianus* were more numerous than in our study of overwintering under *P.* x *hispanica* bark. *Ph. coriaceus* was caught in similar numbers of individuals [47]. Dubovsky et al. [52] studied thrips communities inhabiting tree bark in uneven-aged natural oak forests (*Aceri tatarici-Quercetum*) in Slovakia. They used tree photoeclectors to collect bark-dwelling thrips from oak tree trunks. Their research resulted in the collection of 35 thrips species, of which nine were dendrophilous foliophages, e.g., the most numerous *Thrips minutissimus* Linnaeus and *Mycterothrips albidicornis* (Knechtel), which probably used the trunk as a corridor between the ground and the tree canopy. Those species were not found under the plane tree bark because they overwintered in the soil near the trunks of their host trees. Eight mycophages were trapped on the bark infrequently, similar to the fungivorous species found in our research. The PPCA statistical analysis showed relationships between thrips communities and the oak age. As the trunk circumference was positively correlated with the tree age, the results can be compared with our RDA. In both analyses, species such as *D. ornatus* and *X. fuliginosus* showed a negative correlation with the tree age and the tree trunk circumference, respectively, while the fungivorous species showed a positive correlation [52]. The fungivorous thrips are most commonly found in natural old-growth forests, where they inhabit rotting trunks on the forest floor and feed on a variety of fungal species present under the bark of these trunks [14,15,16,47,52,53,57]. This was confirmed by the positive correlation (RDA) between the presence of these species and the circumference of the studied plane trees. Litavský et al. [24] also found a statistically significant impact of the age of trees in the studied sites on the dispersion of harvestmen species. The negative correlation between the presence of *T. major* and the insolation factor found in our study is due to the fact that this polyphagous species is associated with herbaceous plants. It is most commonly found in shady undergrowth [11,35].

Although winter is a crucial period in animal life, this phase of the life cycle of insects and other arthropods has rarely been studied. Temperature is one of the main factors determining the proper development of insects and the population size in the following spring and summer [59]. A factor that increases insect mortality is the high fluctuation of day and night temperatures, especially in spring, which may be dangerous for insects with soft bodies like thrips larvae and pupae [5,6,7,8,60]. Lewis [60], investigating the overwintering of *Limothrips cerealium* under tree bark, found that thrips mostly die in spring when the bark is too warm and dry and when thrips have started to emerge. Heavy rains or melting snow can kill life stages, e.g., larvae or pupae, overwintering in the soil, while overwintering on trunks partially protects insects from these phenomena. This may be why only adults were collected in our research. On the other hand, Tubulifera species overwintering in scattered aggregations composed of larvae, pupae, and adults may be difficult to find and collect in wintertime [Kucharczyk unpublished data]. However, Rozwałka and Olbrycht [32] found only juvenile forms of several species of spiders hibernating under the bark of plane trees. The specific microhabitats that develop under the overhanging and multi-layered bark characteristic of London trees provide convenient sites for overwintering insects, such as the true bugs (Hemiptera: Heteroptera), which have been studied previously [31]. In 75 sites in western and southeastern Poland, the authors found 25 species, including three invasive species: plane tree bug *Arocatus longiceps*, lime seed bug *Oxycarenus lavaterae* (Fabricius) (Hemiptera: Lygaeidae), and sycamore lace bug *Corythucha ciliata*. Moreover, the authors found that the most common species, *Deraeocoris lutescens* (Schilling) (Hemiptera: Miridae) and *Orius minutus* Wollf (Hemipteran: Anthocoridae), are very widespread in the country, and plane trees should be considered typical sites for their development and overwintering [31].

In comparison with the substantially more numerous order of Hemiptera, finding 15 out of 236 species of thrips known from Poland seems to be a significant number. All thrips species collected under the bark of plane trees are native to Central Europe [35,36]. Some of them, such as polyphagous *T. major*, *T. fuscipennis*, and *F. intonsa*, are associated with herbaceous plants in their life cycle. They feed and reproduce on their leaves or flowers. Among the dendrophilous species found during our research, *D. ornatus* feeds and develops on ornamental shrubs often planted in parks, such as lilacs *Syringa* spp., jasmines (*Philadelphus* spp.) (Oleaceae), and spireas *Spirea* spp. (Rosaceae) [2,35,61]. Particular attention should be paid to fungivorous species, which have rarely been recorded in Poland so far, and their diversity is still not well recognized [55,62]. Due to the limited data, thrips overwintering under tree bark requires further research.

Our research confirmed the role of trees grown in urban and rural parks as habitats for overwintering insects and in preserving the biodiversity of anthropogenically modified areas.

## 5. Conclusions

Our research has shown that the bark of the plane tree (*Platanus* x *hispanica*) can offer good conditions for thrips to overwinter. Thrips found under the bark during winter can be classified as accidental or associated with the tree by feeding on the leaf blades of the tree canopies or fungi growing on the trunk and branches. The studies demonstrated that old trees grown in urban and rural parks are the centers of local diversity where rare and interesting thrips species may be found. The knowledge of insect feeding, reproduction, and overwintering on tree trunks is insufficient and requires further research. Nevertheless, this is the first scientific report on thrips to show the potential of the plane tree bark as an overwintering site for insects of the Thysanoptera order.

## Data Availability

The data presented in this study are available on request from the corresponding author.

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
