# Peer review of "Overwintering of Thrips (Thysanoptera) Under the Bark of the Plane Tree (Platanus x hispanica Mill. ex Münchh.) in Southeastern Poland"

_insects, 2025, doi:10.3390/insects16010092_

Round 1
Reviewer 1 Report
Comments and Suggestions for Authors
The ecological survey investigated overwintering association of thrips with the bark of Platanus x hispanica in Poland. Samples were collected from 29 urban and rural parks. The authors found 15 species of thrips. Effects of insolation, trunk circumference, and the number of trunks were found to influence the distribution dynamics of thrips. The study looks interesting, but I feel that the overall presentation could be improved. My comments are:
L22: I would suggest using rarely studied instead of poorly understood.
L25-26: Revise this sentence for more clarity: Have you sampled it from February to April in all three years or February 2013, March 2014, and April 2015?
L28: It is unclear “Thrips were selected in the laboratory….” Do you want to say that you sorted thrips in the lab?
L32: Replace: The most abundant thrips species included….
L38-40: End with a concluding sentence. This sentence does not fit here. Also, what is meant by wintering thrips? Do you want to say overwintering species of thrips? It is unclear.
L44-46: Give a classical reference here which reported thrips mandibular morphology. For example:
Mound, L. A. (1971). The feeding apparatus of thrips. Bulletin of entomological research, 60(4), 547-548.
L58-59: Revise this sentence: Some species also reproduce through parthenogenesis….
L61-66: Add some references here.
L74: Do you want to say “… are more frequently studied.”
L74: Add family and order.
L83-84: Add reference here.
L89: Add family of the plant.
L114: Be careful about the use of brackets.
L126-129: Add GPS coordinates to identify the sites.
L142: Move this sentence to the first paragraph of materials and methods.
L144: Be careful about the superscripts. For example, m2.
L145: What is meant by selection. I would say, use a better expression to clarify your methodology.
L145: Stereomicroscope make, model and manufacturer.
L160: Use a better heading.
L178: What is the Simpson (1–D) formula?
L204: Instead of writing …. are good places for thrips overwintering, you can write it as provided a suitable habitat for thrips overwintering.
Figure 2: Relabel it. Instead of 1, 2, 3, do A, B, C.
Figure 3 caption: What is a and b. Reflect it in the caption, too.
Table 1: Organize the table as per the MDPI format. Font size and arrangement need attention.
Table 2 and 3: Despite the significant differences between the different treatments, you have not made the simple tables/figures of mean values along with the letters to separate the means. It is important just to see the differences between different treatments.
Discussion: When you are mentioning the species, also mention the order and family. So a full name would look like this:
Bark beetle Cucujus cinnaberinus (Scopoli) (Coleoptera: Cucujidae)
L414-415: You are making a big claim here. Give some references.
L414-420: The style is vague here. Whatever you write should be backed by literature. Also try to be focused on temperature changes. Rewrite this segment for clarity.
L451-452 and other parts where you made big claims:
There have been previous studies available regarding bark dwelling thrips. See following studies:
Lewis, T. (1962). The effects of temperature and relative humidity on mortality in Limothrips cerealium Haliday (Thysanoptera) overwintering in bark. Annals of applied Biology, 50(2), 313-326.
Lewis, T., & Navas, D. E. (1962). Thysanopteran populations overwintering in hedge bottoms, grass litter and bark. Annals of Applied Biology, 50(2), 299-311.
Dubovský, M., & Masarovič, R. (2008). Bark-dwelling thrips (Thysanoptera) and other arthropods in xerothermophilous oak woods in SW Slovakia (preliminary results). Thysanopteron, 3(1).
Tree, D. J., & Walter, G. H. (2012). Diversity and abundance of fungivorous thrips (Thysanoptera) associated with leaf-litter and bark across forest types and two tree genera in subtropical Australia. Journal of Natural History, 46(47-48), 2897-2918.
Author Response
Dear Reviewer,
we have
L22: I would suggest using rarely s3-84: Add reference here.
The references were added.
L89: Add family of the plant.
Name of the family (Platanaceae) was added.
L114: Be careful about the use of brackets.
Thank you for the remarque.
L126-129: Add GPS coordinates to identify the sites.
We prepared the Supplementary materials containing a table with GPS coordinates and dates of thrips collections on particular sites.
L142: Move this sentence to the first paragraph of materials and methods.
This sentence has been moved to the first paragraph.
L144: Be careful about the superscripts. For example, m2.
Thank you for drawing attention to the record.
L145: What is meant by selection. I would say, use a better expression to clarify your methodology.
L148-150:We have clarified how to select thrips from plane tree bark.
L145: Stereomicroscope make, model and manufacturer.
The data has been completed.
Dear Reviewer,
Thank you for all your comments and suggestions. Our responses are included in the added file.

Reviewer 2 Report
Comments and Suggestions for Authors
I have read this paper with interest. Particularly interesting is the breadth of coverage by the authors - historically and ecologically. I am not competent to comment on their statistical analyses, but their work confirms the use of the trunk of non native trees as overwintering sites by thrips in the northern Palaearctic. Line 421 I regard as spurious, because the reason why no Thripidae larvae were found is simply because the species breed on leaves or in flowers. It is interesting that no larvae of the Tubulifera species were found. But I suggest that these species commonly live in aggregated populations that often must be searched for. I look forward to seeing this paper published - the quality of the English is excellent.
Author Response
Dear reviewer,
Thank you for positively accepting our article and your opinion that it is interesting. Of course, the lack of the preimaginal stages in this research may be the other places where they develop and overwinter. In the Discussion, we added information about Tubulifera species based on the first author's observations because references to the overwintering of Tubulifera species are insufficient.
Reviewer 3 Report
Comments and Suggestions for Authors
In this manuscript, The authors conducted a three-year field investigation on species diversity of the overwintering thrips under the bark of the plane tree. They collected and identified the thrips in 29 sites, including urban and rural parks, over three winter seasons from 2013 to 2015. The study identified 15 thrips species and revealed the correlation between thrips and environmental factors. The study is relatively satisfied, but there are many problems that need to be clarified in other aspects, major revision as follows:
1. Thrirps is very small and is difficult to be found in the field. How did the author specifically collect them from the bark of plane three?
2. Were the samples typical and could them represent the sampling site? Did these species significantly changed in different years or different sites?
3. Were all overwintering thrips be adults? Were there larvae collected and what was their radio?
4. Some thrips species were divided into hygrophilous, graminicolous, predatory. The references should be provided.
5. More environmental factors should be added to analysis the correlation.
Comments on the Quality of English LanguageThe manuscript was generally well written.
Author Response
Dear reviewer,
Thank you for all your comments and suggestions. Our responses are included in the added file.

Round 2
Reviewer 1 Report
Comments and Suggestions for Authors
Authors have made a good effort to revise the manuscript. Some minor mistakes:
L93: The family name is never italicized.
L151, 154, 156: Add the manufacturer details. Is it (Olympus Corporation, Tokyo, Japan)?
L244: Acronim? Is it Acronym? (see the heading of second column).
Tables are not left indent. Make them as per MDPI format.
The authors are encouraged to read through the complete manuscript at least twice and make sure that there are no spelling mistakes. Good luck!
Author Response
Dear Reviewer,
Thank you for your recent comments. We have included them in the text. We agree that a family name in zoology is never italicized, but the rule for species names in botany and bacteriology is not clear:
"Family names: There is some confusion about how family names should be written. In American usage, the family name is not usually italicized. The most recent edition of the International Code of Nomenclature for algae, fungi, and plants (the official authority on plant names) recommends that all plant names be in a different font from the rest of the text. The Royal Horticultural Society (U.K.) recommends that family names be italicized. Plant labels in botanical gardens usually have the family name in capital letters, for instance, PINACEAE" - it is why the Platanaceae name was written in italics.
The table was changed but the Editor suggested to retain the current formatting.
Reviewer 3 Report
Comments and Suggestions for Authors
Accept for pubulifcation in present form.
Author Response
Dear Reviewer,
Thank you for accepting our changes in the manuscript; your remarks helped us prepare it better.